# Robust automated reading of the skin prick test via 3D imaging and parametric surface fitting

Jesus Pineda[1], Raul Vargas[1], Lenny A. Romero[2], Javier Marrugo[3], Jaime Meneses[4], Andres G. Marrugo [1] *

1 Facultad de Ingeniería, Universidad Tecnologica de Bolivar, Cartagena, Colombia, 2 Facultad de Ciencias Básicas, Universidad Tecnologica de Bolivar, Cartagena, Colombia, 3 Instituto de Investigaciones Inmunológicas, Universidad De Cartagena, Cartagena, Colombia, 4 Grupo de Óptica y Tratamiento de Señales, Universidad Industrial de Santander, Bucaramanga, Colombia

* agmarrugo@utb.edu.co

**Data Availability Statement:** The authors confirm that all data underlying the findings are fully available without restriction. The final dataset and accompanying code are available on the Open

## Abstract

The conventional reading of the skin prick test (SPT) for diagnosing allergies is prone to inter- and intra-observer variations. Drawing the contours of the skin wheals from the SPT and scanning them for computer processing is cumbersome. However, 3D scanning technology promises the best results in terms of accuracy, fast acquisition, and processing. In this work, we present a wide-field 3D imaging system for the 3D reconstruction of the SPT, and we propose an automated method for the measurement of the skin wheals. The automated measurement is based on pyramidal decomposition and parametric 3D surface fitting for estimating the sizes of the wheals directly. We proposed two parametric models for the diameter estimation. Model 1 is based on an inverted Elliptical Paraboloid function, and model 2 on a super-Gaussian function. The accuracy of the 3D imaging system was evaluated with validation objects obtaining transversal and depth accuracies within ± 0.1 mm and ± 0.01 mm, respectively. We tested the method on 80 SPTs conducted in volunteer subjects, which resulted in 61 detected wheals. We analyzed the accuracy of the models against manual reference measurements from a physician and obtained that the parametric model 2 on average yields diameters closer to the reference measurements (model 1: -0.398 mm vs. model 2: -0.339 mm) with narrower 95% limits of agreement (model 1: [-1.58, 0.78] mm vs. model 2: [-1.39, 0.71] mm) in a Bland-Altman analysis. In one subject, we tested the reproducibility of the method by registering the forearm under five different poses obtaining a maximum coefficient of variation of 5.24% in the estimated wheal diameters. The proposed method delivers accurate and reproducible measurements of the SPT.

## Introduction

The skin prick test (SPT) is the most commonly used method for diagnosing asthma, allergic rhinitis, and food allergies [1]. It is relatively simple and quick to read, and reproduces allergic reactions by type I hypersensitivity. These health conditions affect an estimated 30% of the

Science Framework DOI https://doi.org/10.17605/OSF.IO/YK8U5.

**Funding:** This study was supported by Colciencias (www.colciencias.gov.co, Grant 538871552485) and by Universidad Tecnológica de Bolívar (www.utb.edu.co, Grants C2018P005 and C2018P018), Colombia.

**Competing interests:** AGM, LAR, J. Marrugo and J, Meneses are co-inventors of a patent application related to this study, Dispositivo y método de reconstrucción 3D para la medición de pápulas en la piel. Colombian Patent App. NC2018/0007546; Inventors: Andres G. Marrugo, Lenny A. Romero, Javier Marrugo, Jaime Meneses. Assignee: Universidad Tecnologica de Bolivar, Universidad de Cartagena, Universidad Industrial de Santander. There are no further patents, products in development or marketed products to declare. This does not alter our adherence to all the PLOS ONE policies on sharing data and materials, as detailed online in the guide for authors.

world population [2–4] with incidence on the rise. Hence, the need for continuous optimization of related diagnostic tools and therapies [5].

In the SPT, several allergens are introduced into the skin of the patient simultaneously. The SPT is carried out either by placing drops of the allergens on the skin and pricking with a lancet, or with a multi-test device, as the one shown in Fig 1(A). A small swelling of the skin called wheal or papule appears when there is a reaction. The size of the wheal determines the degree of sensitization [6]. For routine and most study settings, the wheals are measured directly on the skin by a physician with a millimeter-graded ruler as shown in Fig 1(B).

The shapes of the wheals vary considerably, which complicates the SPT assessment [7], as illustrated in Fig 1(C). It is generally accepted that the most accurate way to assess a wheal response is planimetry from a traced copy, also called the "scanned area" method. However, it is difficult to perform manually and time consuming [8–10]. To simplify and expedite the SPT assessment the wheal shape is commonly characterized by an average diameter assuming the wheal may be described reasonably well by an ellipse with the longest diameter $d_{max}$ and the orthogonal midpoint diameter $d_{min}$, as shown in Fig 1(D). However, this approximation and other similar methods have been shown prone to errors [7], and have lead to diminished comparability when SPT results are reported [11]. To standardize the SPT assessment, the Global Allergy and Asthma European Network laid guidelines, which include a standard protocol for SPT assessment [11]. Among other recommendations, it states that only the largest diameter of the wheal of each test is measured, a positive being of $\geq 3$ mm. The reason is that the longest diameter is a better estimate of wheal surface area than the mean perpendicular diameters [9]. This assessment is illustrated in Fig 1(D) with the measurement of $d_{max}$ regardless of trying to approximate the wheal shape by an ellipse.

Wheals fade quickly, which complicates their assessment and documentation. An alternative to the scanned area method to overcome this problem is by taking digital photographs of the test. However, the visual interpretation of these images produces a significant amount of

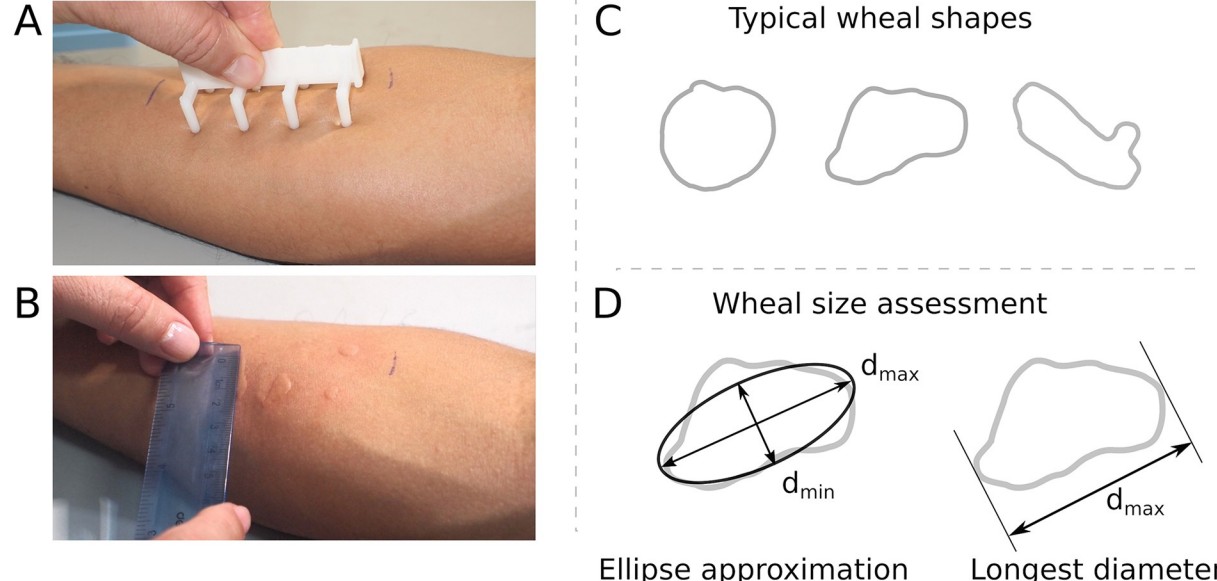

**Fig 1.** **(A)** The physician performs the SPTs with a disposable multi-test on the surface of the forearm. **(B)** After 15 minutes, the physician measures the skin reactions or wheals using a ruler. **(C)** Typical shapes of wheals, from regular to irregular shape with pseudopod. **(D)** The wheal size assessment is approximated as an ellipse or by measuring the longest diameter.

variation [12]. Even assessing the images through digital image processing programs is not entirely reliable and has not been sufficiently studied on different skin tones [13, 14].

Furthermore, the fact that wheals are local elevations of the skin suggests that 3D information of the skin surface should be used as a means to measure and assess the SPT. In a recent review paper by Justo et al. [15] on the evolution towards automated reading of the SPT, the authors state that most of the existing systems for assessing the SPT are not practical for use in a busy clinical practice mostly due to the time needed to obtain the results and the lack of sufficient precision. They concluded that 3D scanners may pave the way to a reliable wheal measurement method. Although, the technology still has to deal with several challenges: simultaneous measurement of multiple wheals, fast computer processing of data and the development of a simple and low-cost device. In this work, we propose a system that tackles these challenges.

3D scanning technology is becoming an essential tool in many medical subfields [16]. It has been applied for the monitoring of skin treatment therapy [17], or the characterization of cancer lesions [18]. However, each procedure or application has specific metrological requirements, like the field of view or the depth resolution, which typically render a device unsuitable for multiple applications. We know of three earlier works which make use of 3D technology for assessing the SPT [19–21]. The work of dos Santos et al. [20] was the first to show the advantages of 3D imaging technology for measuring wheals. Although, they used a commercial 3D scanning device [22, 23] with a relatively small field of view (FOV) in the order of 40×30 mm, which could only register one wheal at a time. This limitation impedes its use in the typical clinical setting where many SPTs are typically applied simultaneously. In a feasibility study by Verdaasdonk et al. [21], the authors used general-purpose 3D scanners for assessing the SPT. They tested the scanners extensively on phantom objects, like buttons, but provided few clinical data (17 positive allergic reactions) which were assigned a reaction grade 1, 2, or 3 by a dermatologist instead of validating with a more quantitative approach. Also, they did not discuss any automated detection and measurement procedure.

In the recent work by Justo et al. [19], they use a 3D laser scanner system that registers the whole surface of the forearm, making it more suitable for real clinical situations. It is based on the laser-line triangulation principle by moving a scanning head along the forearm of the patient where the SPT was applied. However, their device does not capture the texture image of the skin, which may be useful for assessing other aspects of the SPT like the erythema. Also, they report that their method is prone to detect false positives (from 169 detections only 97 where true wheals). The authors test their device extensively against validation objects, but they do not report measurement results or accuracy assessment from real wheals from SPTs. Moreover, the fact that it requires a moving scanning head makes the device prone to patient movement.

## Contribution

In this work, we use an in-house developed wide-field 3D imaging system [24], shown in Fig 2, for the 3D reconstruction of the SPT, which we validate with objects of known shape shown in Fig 3. We propose a framework for the robust automated measurement of the skin wheals. Our proposed method, depicted in Fig 4, is based on advanced 3D data processing. We perform a pyramidal decomposition of the 3D reconstruction to filter out the noise and the global surface of the arm, leaving only the 3D reconstruction of the wheals. The automated measurement is based on a robust parametric fitting to the 3D data of the wheals for obtaining the longest diameter directly. Although it could also be used to estimate the wheal area or volume, for this work we chose the longest diameter because it is extensively used in the clinical practice,

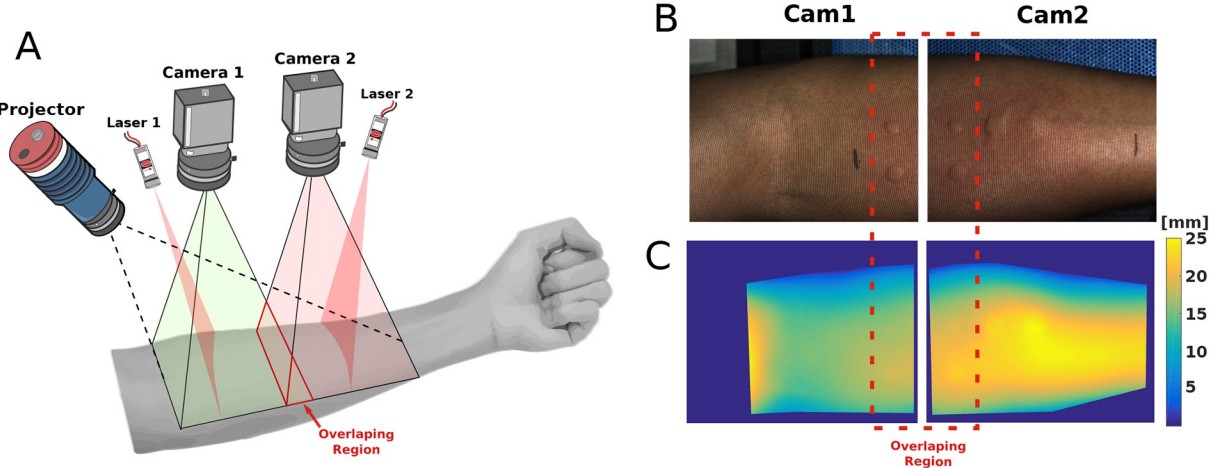

**Fig 2. (A)** 3D imaging system. **(B)** Captured fringe images. **(C)** 3D Reconstructions. The dotted red line denotes the overlapping region between the observation systems.

and the results are readily interpretable. This approach is a step forward in bringing 3D parametric models to characterizing skin features. Dimensional metrology of anatomical features is difficult; there are no two anatomical features alike. However, that has not prevented the development of anatomical parametric models which work for the majority of the population [25], and with few numbers allow us to describe or match two similar features. In the same way, our model attempts to generalize the irregular shape of the skin wheal by taking into account the 3D shape to produce an accurate and reliable measurement. To the best of our knowledge, this is the first time such a model is proposed for the measurement of the wheals in the SPT. In the following sections, we describe the 3D imaging system, the wheal detection and measurement approach, the methods, the experimental results, the discussion, and finally our concluding remarks.

## 3D Imaging system

A schematic of the 3D imaging system [24, 26] that we designed is shown in Fig 2A. It is a wide-field 3D fringe projection system capable of reconstructing large areas ($\sim$ 150 mm × 250 mm), e.g., the whole surface of the forearm, with a height accuracy in the order of 0.01 mm. It consists of three major parts: an acquisition system, a projection system, and a control unit.

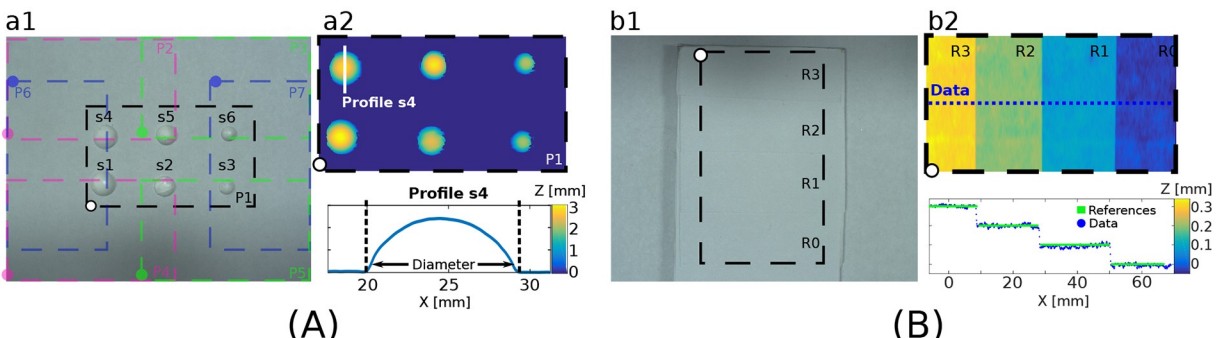

**Fig 3. Validation experiments: (A)** measuring spherical caps of known diameters and **(B)** steps with known height.

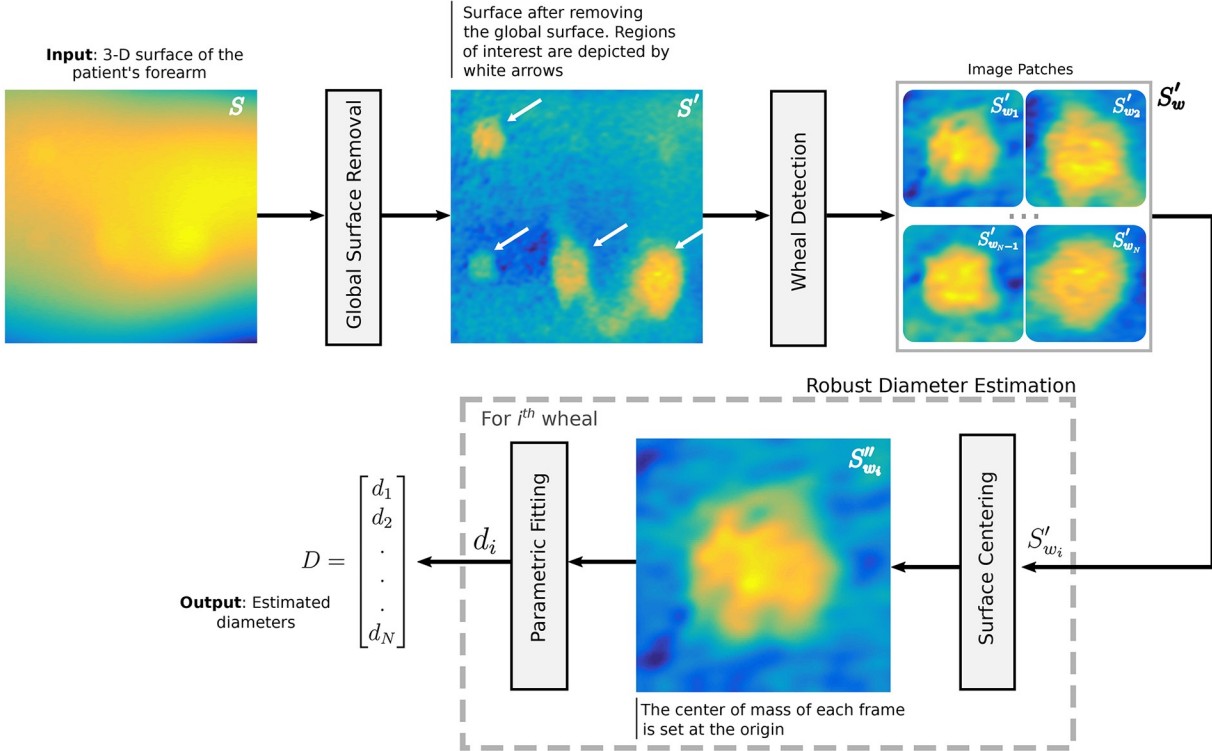

**Fig 4. Block diagram illustrating the proposed method for the measurement of skin wheals.** $S$ and $D$ denote the 3D surface of the patient's forearm and the estimated diameters, respectively. The other variables are intermediate outputs of every stage; their meaning is given in the text.

The acquisition system consists of two color cameras Basler Ace 1300-60gc with 16 mm focal length lenses (Computar M1614-MP2) at F/1.4, with a resolution of 1280 × 1024, and a maximum frame rate of 60 frames/s. The projection system is comprised by an LED pattern projector (Optoengineering LTPRHP3W-W) that contains a stripe pattern of 400 lines with line thickness 0.01 mm with a projection lens of 12 mm focal length (Edmund Optics 58001), and two laser line projectors (SYD1230) with wavelength 650 nm. The control unit consists mainly of a computer that controls the acquisition, and the projection devices. The lasers allow precise positioning of the forearm beneath the 3D imaging device. When the forearm is correctly positioned, each camera acquires a laser line image, a fringe image, and a texture image, as shown in Fig 2(B). The acquisition is done in less than 500 ms. Note that there is a small overlapping region for redundancy and a smoother transition from one 3D reconstruction to the other. The fringe images are processed independently to obtain two 3D reconstructions via Fourier Transform Profilometry [27, 28], as shown in Fig 2(C). The 3D reconstructions are automatically merged in a global coordinate system through a previous calibration [29]. Notice that the wheals are included in the topography, but to measure them accurately the global surface has to be removed. For further details of the 3D imaging system, see Ref. [24].

## 3D System resolution and accuracy

To evaluate the resolution and accuracy of the 3D imaging system, we carried out two validation experiments. The first experiment consisted of reconstructing six spherical caps (s1, . . ., s6) of different diameters, as shown in Fig 3(A). The diameter of each spherical cap was measured three times with a caliper to the nearest 0.01 mm to obtain the average reference

**Table 1. First validation experiment.** Measured diameters for six spherical caps with a caliper (Reference measurement) and the 3D imaging system (using camera 1 and camera 2). All values in millimeters.

| Spherical cap | Reference | Camera 1 $\bar{x} \pm s$ | Camera 2 $\bar{x} \pm s$ |
|---|---|---|---|
| s1 | 9.08 ± 0.01 | 9.04 ± 0.05 | 9.02 ± 0.08 |
| s2 | 7.42 ± 0.01 | 7.42 ± 0.03 | 7.53 ± 0.10 |
| s3 | 5.65 ± 0.01 | 5.65 ± 0.11 | 5.65 ± 0.08 |
| s4 | 8.88 ± 0.01 | 8.87 ± 0.07 | 8.85 ± 0.09 |
| s5 | 7.42 ± 0.01 | 7.42 ± 0.09 | 7.48 ± 0.05 |
| s6 | 5.50 ± 0.01 | 5.52 ± 0.09 | 5.53 ± 0.09 |

measurement reported in Table 1. Then, we measured the diameter of the spherical caps with the 3D imaging system. We placed them in seven different positions (P1, . . ., P7) throughout the FOV of each camera, as shown in Fig 3(A1). The 3D reconstruction of the spherical caps and a height profile across s4 are shown in Fig 3(A2). In Table 1, we report the values of the average diameter $\bar{x}$ and the standard deviation $s$ of the measurements for each spherical cap obtained for camera 1 and camera 2. The measurements are accurate within ±0.1 mm.

The second validation experiment, shown in Fig 3(B), consisted of measuring an object with known depth steps. We made a staircase with four steps using adhesive paper cutouts with a thickness of 0.100 mm. The tested object is shown in Fig 3(B1), where R0, R1, R2, and R3 correspond to the steps. The height difference between each consecutive step is 0.100 mm. In Fig 3(B2) we show the depth map of the steps and a height profile across the steps obtained with the 3D imaging system. The distances measured between each step and the reference step R0 are reported in Table 2 for each camera. The results show that the 3D system can accurately measure objects with height in the order of 0.100 mm. The height measurements are accurate within ±0.01 mm.

## Wheal detection and measurement

An overview of the proposed method for the measurement of skin wheals is shown in Fig 4. The input to the work-flow is the 3D surface of the patient's forearm $S(x, y, z)$. As the wheals are spread out over the global curvature of the forearm, a *Global Surface Removal* stage is carried out to obtain $S'$ and isolating the wheals. $S'$ includes bumps due to the wheals but also due to skin structural changes caused by pores, follicles, wrinkles, and hair. Therefore, we detect wheals from $S'$ and extract $N$ surface patches referred to as $S'_{w_i}$, where $i \in \{1, 2, . . ., N\}$. For each surface $S'_{w_i}$, we set the origin at the center of mass to obtain $S''_{w_i}$ by means of Principal Component Analysis (PCA) [30]. Finally, we employ a parametric fitting stage for diameter estimation. $D = [d_1, d_2, . . ., d_N]$ and $f$ denote the estimated diameters and the fitted surfaces, respectively.

## Global surface removal

The Global Surface Removal stage in Fig 4 relies on a standard Gaussian-Laplacian pyramid [31] to obtain $S'$. The proposed approach for this stage is illustrated in Fig 5. The input to this

**Table 2. Second validation experiment.** Measured height for three steps from a reference step R0, using the camera 1 and the camera 2. All values in millimeters.

| | Reference | Camera 1 $\bar{x} \pm s$ | Camera 2 $\bar{x} \pm s$ |
|---|---|---|---|
| R0-R1 | 0.100 | 0.099 ± 0.011 | 0.106 ± 0.009 |
| R0-R2 | 0.200 | 0.204 ± 0.011 | 0.204 ± 0.008 |
| R0-R3 | 0.300 | 0.300 ± 0.012 | 0.304 ± 0.008 |

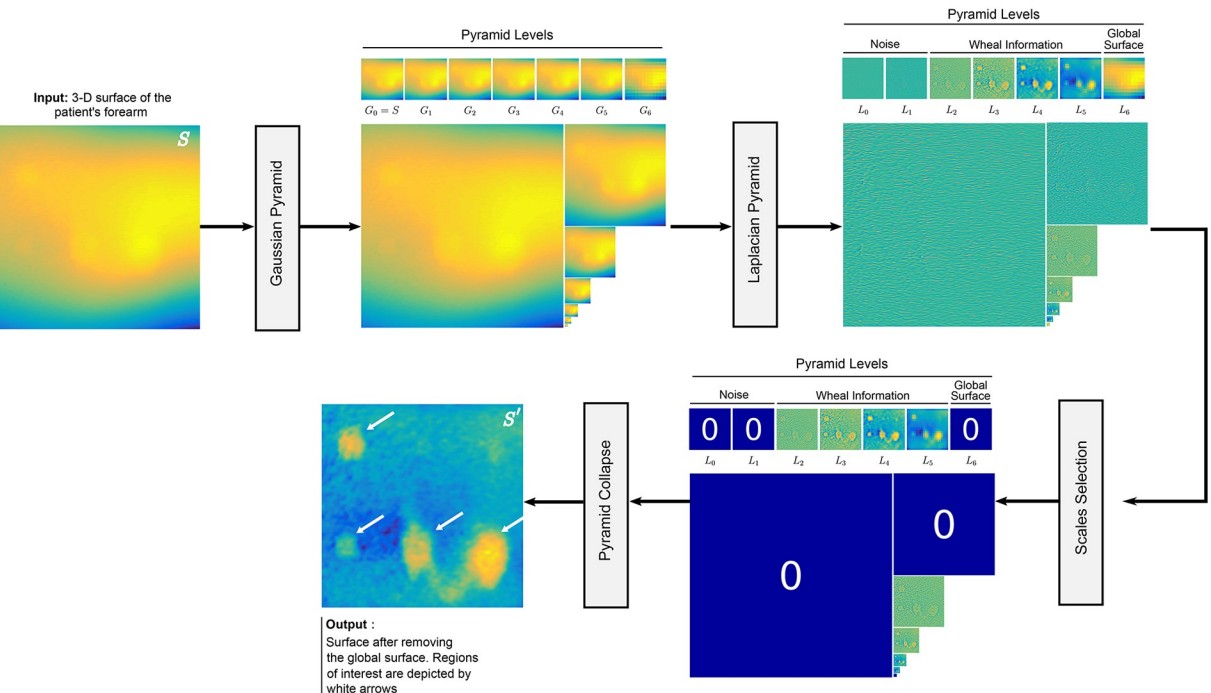

**Fig 5. The *Global Surface Removal* stage.** $S$ and $S'$ denote the 3D surface of the patient's forearm and the surface after removing the global curvature of the forearm, respectively. The other variables are intermediate outputs of every sub-stage; their meaning is given in the text.

stage is the 3D surface of the patient's arm $S$. A Gaussian pyramid decomposes $S$ into subsets of progressively lower resolution image versions $G_\ell$ called levels. The original image $G_0 = S$ is convolved by a Gaussian low pass filter. The resulting convolved image is down-sampled to half the width and height as $G_1 = (G_0 * g) \downarrow 2$, where $*$ denotes the convolution operator, $g$ represents the Gaussian kernel function, and $\downarrow 2$ down-samples the image by a factor of 2. This process is iterated $n$ times for each level of the Gaussian pyramid until $G_n$ has only a few pixels. As shown in Fig 5, we use $n = 7$ for the surface decomposition.

The Laplacian pyramid is obtained by differencing the image at level $\ell$ and at its approximation at the following coarser scale of the Gaussian pyramid as $L_\ell = G_\ell - (G_{\ell+1} \uparrow 2) * h$, where $\uparrow 2$ up-samples $G_{\ell+1}$ by doubling its size, and $h$ is a smoothing kernel. Each level of the Laplacian pyramid $L_\ell$ represents details that distinguish successive levels of the Gaussian pyramid [32, 33]. Note in Fig 5, the two lower levels of the Laplacian pyramid, i.e., $\ell \in \{0, 1\}$, are associated with noise. Similarly, the 3D shape of a wheal is mostly contained in levels $\ell \in \{2, 3, 4, 5\}$. The highest level of the Laplacian pyramid is defined as $L_n = G_n$ and represents the global curvature of the arm. Finally, by zeroing out the levels not related to the 3D information of the wheals, the Laplacian pyramid is collapsed to obtain $S'$, i.e., the 3D information after removing the global surface, as shown in Fig 5. The wheals are now easily identifiable.

## Wheal detection

The block diagram in Fig 6 illustrates the *Wheal Detection* stage. Since the wheals can be broadly modeled as image blobs, this stage relies on a multi-scale Laplacian of Gaussian (LoG) blob detector to automatically detect the wheals [34]. The input to the method is the surface $S'$. We compute a scale-space representation from $S'$ through Laplacian of Gaussian filters with successively increasing standard deviations $\sigma$. This scale-space representation may be regarded

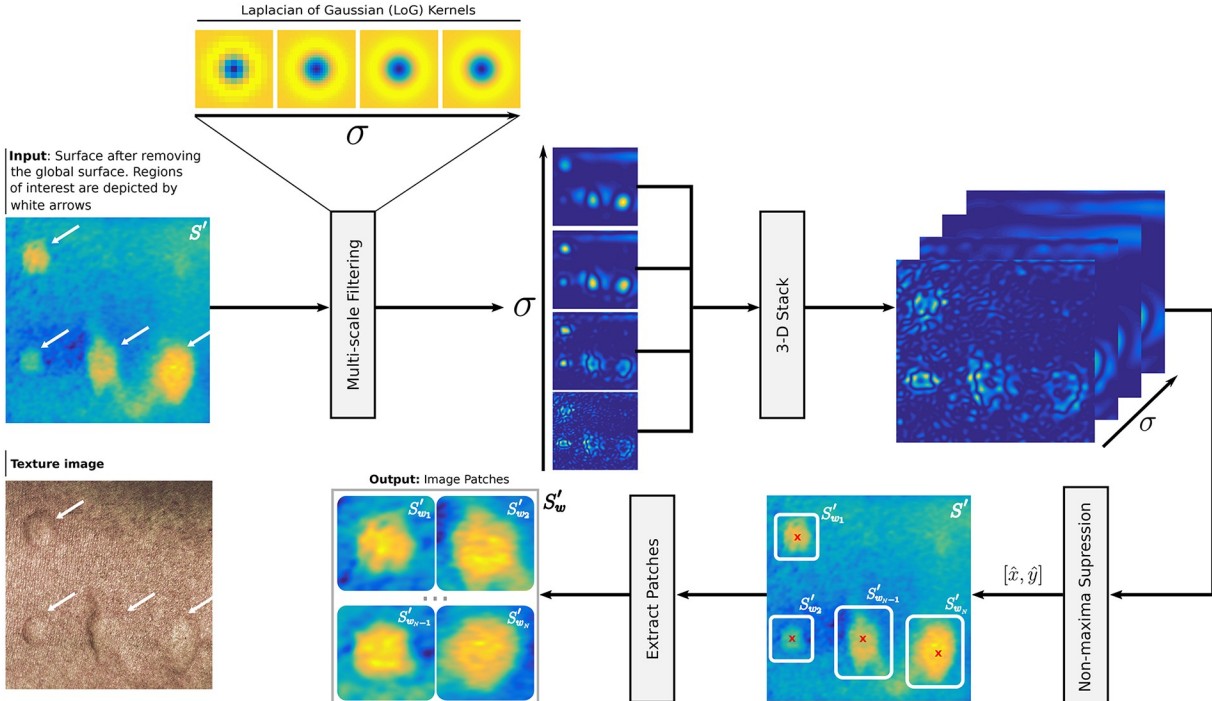

**Fig 6. The *Wheal Detection* stage.** $S'$ and $S'_w$ denote the surface after removing the global curvature of the forearm and the extracted surface patches, respectively. We have included the texture image to highlight the correct detection of the wheals. This image has been digitally sharpened to improve the visualization of the wheals.

as a 3D stack of filtered versions of $S'$. Then, a non-maxima suppression is carried out to detect scale-space maxima. This sub-stage returns the coordinates at which the maxima were detected. Finally, based on this information, we extract $S'_w$.

## Robust diameter estimation via parametric fitting

The purpose of this stage is to fit a parametric model that automatically estimates a clinically relevant parameter that is readily interpretable and comparable to the standard reading of the SPT. In other words, we are trying to mimic computationally what the physician does with a ruler, but doing so in a way that produces accurate and consistent results every time the same wheal is measured.

We define a general parametric fitting framework, and we test two parametric models inspired by the two conventional wheal measurement procedures described in the introduction. With the parametric model 1, we attempt to reproduce the shape of the wheal with an inverted Elliptical Paraboloid function, such that when evaluated at $z = 0$ yields an ellipse. The major axis of the ellipse is adopted as the longest wheal diameter. With the parametric model 2, we approximate the longest wheal diameter as the longest dimension of a bounding box. Bounding boxes of probabilistic nature or confidence score have been massively used for object detection algorithms [35], and we believe this is a reasonable approach since wheals often have irregular shapes. Furthermore, since wheals have been reported as having a relatively flat top [19] and decreasing smoothly, we use a generalized version of the Gaussian function called the super-Gaussian which has more degrees of freedom. The super-Gaussian distribution has been used to model the intensity distribution of laser beams and determining

its width [36], or for determining the size of micro-aneurysms [37], and other distributions [38].

The parametric fitting stage consists of three steps:

1. For each surface $S'_{w_i}$, we set the origin at the center of mass to obtain $S''_{w_i}$ by means of PCA [30].

2. From $S''_{w_i}$, we fit either of the two parametric models to the 3D surface data. To incorporate a rotational degree of freedom, the fitting is carried out in an $(x', y')$-coordinate system rotated clockwise about an angle $\theta$ given by

$$\begin{bmatrix} x' \\ y' \end{bmatrix} = \begin{bmatrix} \cos\theta & -\sin\theta \\ \sin\theta & \cos\theta \end{bmatrix} \begin{bmatrix} x \\ y \end{bmatrix}. \tag{1}$$

The angle $\theta$ now becomes an additional parameter to be estimated as part of each model. The parametric model optimization is solved with the Trust-region reflective non-linear least-squares algorithm [39].

3. Once the optimization is complete, the obtained relevant parameter per model yields the estimation of the wheal longest diameter.

**Parametric model 1.** The parametric model 1 is based on an inverted Elliptical Parabaloid function, given by,

$$f_1(x', y') = -\frac{1}{c}\left(\frac{(x'-x'_0)^2}{w_{x'}^2} + \frac{(y'-y'_0)^2}{w_{y'}^2}\right) + \beta, \tag{2}$$

where $w_{x'}$ and $w_{y'}$ model the width of the wheal in $x'$- and $y'$-dimension, respectively; $\beta$ is the surface offset; and $c$ is parameter modeling the wheal height. A mathematical expression for the diameter of the wheal is obtained by evaluating $f_1 = 0$, as follows,

$$\left(\frac{(x'-x'_0)^2}{a^2} + \frac{(y'-y'_0)^2}{b^2}\right) = 1, \tag{3}$$

where $a = \sqrt{\beta c w_{x'}^2}$ and $b = \sqrt{\beta c w_{y'}^2}$ are the relevant parameters for this model. Finally, the diameter $d_i$ is computed as $d_i = \max\{2a, 2b\}$, i.e., the longest diameter, as is customary for the standard skin prick test reading [9].

**Parametric model 2.** In the parametric model 2, the skin wheals are modeled as 2D Super-Gaussian functions, given by,

$$f_2(x', y') = \gamma * \exp\left(-\left(\frac{(x'-x'_0)^2}{2\sigma_{x'}^2}\right)^{\beta_{x'}} - \left(\frac{(y'-y'_0)^2}{2\sigma_{y'}^2}\right)^{\beta_{y'}}\right), \tag{4}$$

where $\gamma$ is the parameter modeling the wheal height; $\beta_{x'}$ and $\beta_{y'}$ model the wheal flatness in $x'$- and $y'$-dimension, respectively; $\sigma_{x'}$ and $\sigma_{y'}$ are the relevant parameters modeling the width of the wheal in $x'$- and $y'$-dimension, respectively. By fitting Eq (4) to $S''_{w_i}$, the measured diameter $d_i$ is computed as $d_i = \max\{4\sigma_{x'_i}, 4\sigma_{y'_i}\}$, which exhibits good agreement with the spatial distribution of the wheal.

## Materials and methods

### Subjects and tests

Subjects in the age of 18-60 years suspected of having an allergic reaction against inhalant allergens were eligible for enrollment in the study. In total, 7 females and 2 males (mean age: 24.11, range: 18–48 years) were enrolled. All the subjects had to sign the declaration of consent before participating in the study. The study protocol was approved by the ethics committee of the Universidad Tecnológica de Bolívar, Colombia.

A physician performed the SPT on the subjects by applying an array of 6 allergens, negative and positive control (Histamine, 10mg/ml) using a sterile disposable Multi-test (Multi-TestRPC, Lincoln Diagnostics, Inc, Decatur, IL. USA) on the volar surface of the forearm. Extracts of common allergens (Inmunotek, Madrid, Spain) were applied including: *Blomia tropicalis*, *Dermatophagoides pteronyssinus*, *Dermatophagoides farinae*, *Cat dander*, *Periplaneta americana*, and *Dog dander*. A *Positive control* solution (Histamine), and a *Negative control* solution (Diluent) were also applied to the subjects, for a total of 8 SPTs per subject. One subject was skin prick tested on the left and right forearms. Summing up, there were 80 SPTs in the entire study.

After 15 minutes of the application of the SPT on the subjects, the physician measured the skin reactions using the traditional ruler-based method. He reported the measurements of 50 wheals for all subjects. Next, the SPTs for all subjects were digitally registered using the 3D imaging system, as shown in Fig 2, from which 61 wheals were detected. The difference between the wheal measurements reported by the physician (50) and the wheals detected by the 3D imaging system (61) resulted from small wheals that the physician discarded or failed to measure due to their size. However, after the wheals where digitized in 3D, the physician acknowledged them as true wheals because they appeared at points consistent with the prick locations from the multi-test. We highlight that our system has sufficient depth sensitivity to detect small wheals, which would otherwise be discarded through visual inspection.

### Experiments and performance assessment

We designed two experiments to evaluate the suitability of the wheal measurement method over a typical usage scenario. In the first experiment, we evaluate the accuracy and precision of the methods compared to the conventional ruler-based measurement method. In the second experiment, we assess the performance of the methods in terms of reproducibility, i.e., the degree of agreement between several measurements produced under slightly different conditions.

**Experiment 1: Accuracy.** To assess the accuracy of the proposed methods, we created manual reference measurements from the 3D reconstruction of the wheals, a similar approach previously used for the measurement of SPTs [8]. With the help of a custom-built user interface (Fig 7), the physician measured the longest diameter of all SPT reactions. The reference measurement for each wheal was obtained from the mean of three independent measurements. The physician measured the longest diameter for each wheal with the aid of the viewing window, a color-bar indicating height distribution, and a draggable and resizable line, as shown in Fig 7. The wheals were measured sequentially in a loop until each wheal was measured three times. This procedure was carried out for the 61 SPT reactions.

To determine the agreement between the measured wheal diameters and the estimated wheal diameters by the proposed methods, we computed the ratio between the two measurements. A good agreement should yield a mean ratio close to 1. Also, we performed Pearson's two-sided test to evaluate the correlation. However, since correlation alone does not

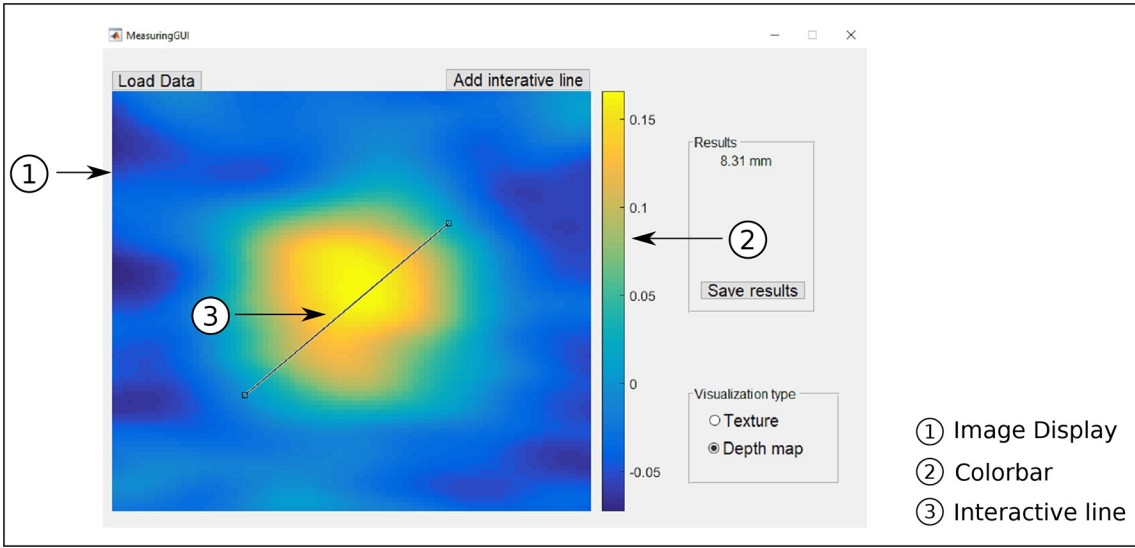

**Fig 7. Custom user interface for the manual reference measurement of the wheal diameter.** Depth colorbar units in mm.

necessarily imply a good agreement between the two methods, we used Bland–Altman plots to assess the agreement [40, 41]. More specifically, to analyze the mean difference $\bar{d}$ between the two methods and to determine the limits of agreement. The Bland–Altman plot displays the difference between the two methods with respect to the best estimate of the true value, i.e., the average of the two measurements. For normally distributed differences, we expect 95% of the differences to lie between $\bar{d} + 1.96$ SD and $\bar{d} - 1.96$ SD. Assuming the manual measurements as the reference method, the limits of agreement can be used as a measure of the total error of the proposed method [42]. In this case, the expected limits of agreement are in the order of ±1 mm. We carried out the same analysis for assessing the agreement between the reference measured wheal diameters and the ruler-based measurements made by the physician.

**Experiment 2: Reproducibility.**   For this experiment, we asked one of the subjects to place his forearm in 5 different positions across the FOV of the 3D imaging system. Each position yields an independent measurement acquired under different conditions, mainly because the skin is not a rigid object. Nevertheless, by moving the arm to a different position and repeating the measurement provides the best evidence of the reproducibility and robustness of the wheal measurement method in a real scenario. Depending on the location of the forearm during acquisition, some wheals were imaged either from both cameras in the overlapping region or only from camera 1 or camera 2. We computed the coefficient of variation (CV) for each wheal from the estimated diameter for each forearm position. The CV is defined as the ratio of the unbiased standard deviation (SD) and the arithmetic mean (CV = SD/mean ×100). It provides statistical information on the dispersion of the measurements. Low CV values are associated with high reproducibility of the wheal measurement method.

## Results

### Experiment 1

**Accuracy of the proposed methods.**   The ratio of the estimated diameters by the two parametric models was close to 1, exactly 0.99 ± 0.022 or (0.97, 1.01) with 99% confidence. The CV of the computed ratios was 6.40%, which exhibits high reproducibility between the

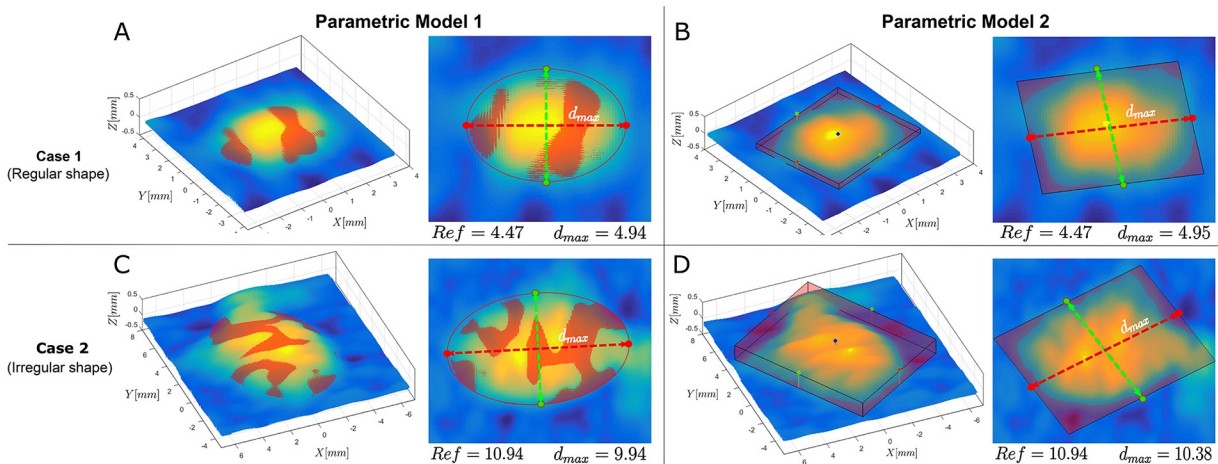

**Fig 8.** The results of fitting a regular-shaped wheal with (**A**) the parametric model 1 and (**B**) the parametric model 2. Both models are in numerical agreement and slightly overestimate the reference measurement of 4.47 mm. For an irregular-shaped wheal the result from (**C**) parametric model 1 and (**D**) parametric model 2 differ by 0.44 mm. However, the result from parametric model 2 results in a more accurate estimation of the longest diameter.

estimations from both methods. To illustrate the differences between the two parametric models, in Fig 8, we show the results on fitting the two models to two different shaped wheals. In Case 1 of Fig 8, we show the results on fitting a regular-shaped wheal. The parametric model 1 produces an estimated diameter of $d_{max}$ = 4.94 mm, whereas the parametric model 2 a $d_{max}$ = 4.95 mm. Both models are in numerical agreement and slightly overestimate the reference measurement of 4.47 mm. In Case 2 of Fig 8, for an irregular-shaped wheal, the estimated diameters from both models differ by 0.44 mm. However, the result from parametric model 2 is closer to the reference. On the one hand, the parametric model 1 is forcing an elliptical shape resulting in parts of the wheal lying outside the ellipse. On the other hand, the parametric model 2 tries to fit the 3D bounding box to the irregular-shaped wheal, which results in a more accurate estimation of the longest diameter.

The ratio of the manual reference measurements and the estimated diameters by the parametric model 1 was 0.93 ± 0.030 or (0.90, 0.96) with 99% confidence. The CV of the computed ratios was 9.62%. Additionally, Pearson's two-sided test shows that the manual references and the estimated diameters were strongly correlated, with Pearson's correlation coefficient $r$ = 0.986 [95% confidence interval, CI = 0.977-0.992, P $\ll$ 0.001]. The regression equation $y$ = 0.913(95% CI = 0.873-0.953)$x$ + 0.971(95% CI = 0.673-1.269) indicates a good agreement between the two measurements. The Bland-Altman analysis confirms a good agreement, with arithmetic mean $\bar{d} = -0.398$ mm and SD = ±0.603 mm. The 95% agreement limits are [−1.58, 0.78] mm.

The ratio of the manual reference measurements and the estimated diameters by the parametric model 2 was was 0.94 ± 0.029 or (0.91-0.97) with 99% confidence. The CV of the computed ratios was 9.02%. The regression equation $y$ = 0.945(95% CI = 0.907-0.984)$x$ + 0.700 (95% CI = 0.414-0.986) and the Pearson's correlation coefficient $r$ = 0.988 [95% confidence interval, CI = 0.980-0.993, P $\ll$ 0.001] indicate a higher correlation than the obtained with the parametric model 1. Additionally, the Bland-Altman analysis further confirms a good agreement between both measurements with arithmetic mean $\bar{d} = -0.339$ mm and SD = ±0.537 mm. The mean difference $\bar{d}$ is not zero, which implies that on average the proposed method measures 0.330 mm more than the manual reference. However, this difference is largely due to

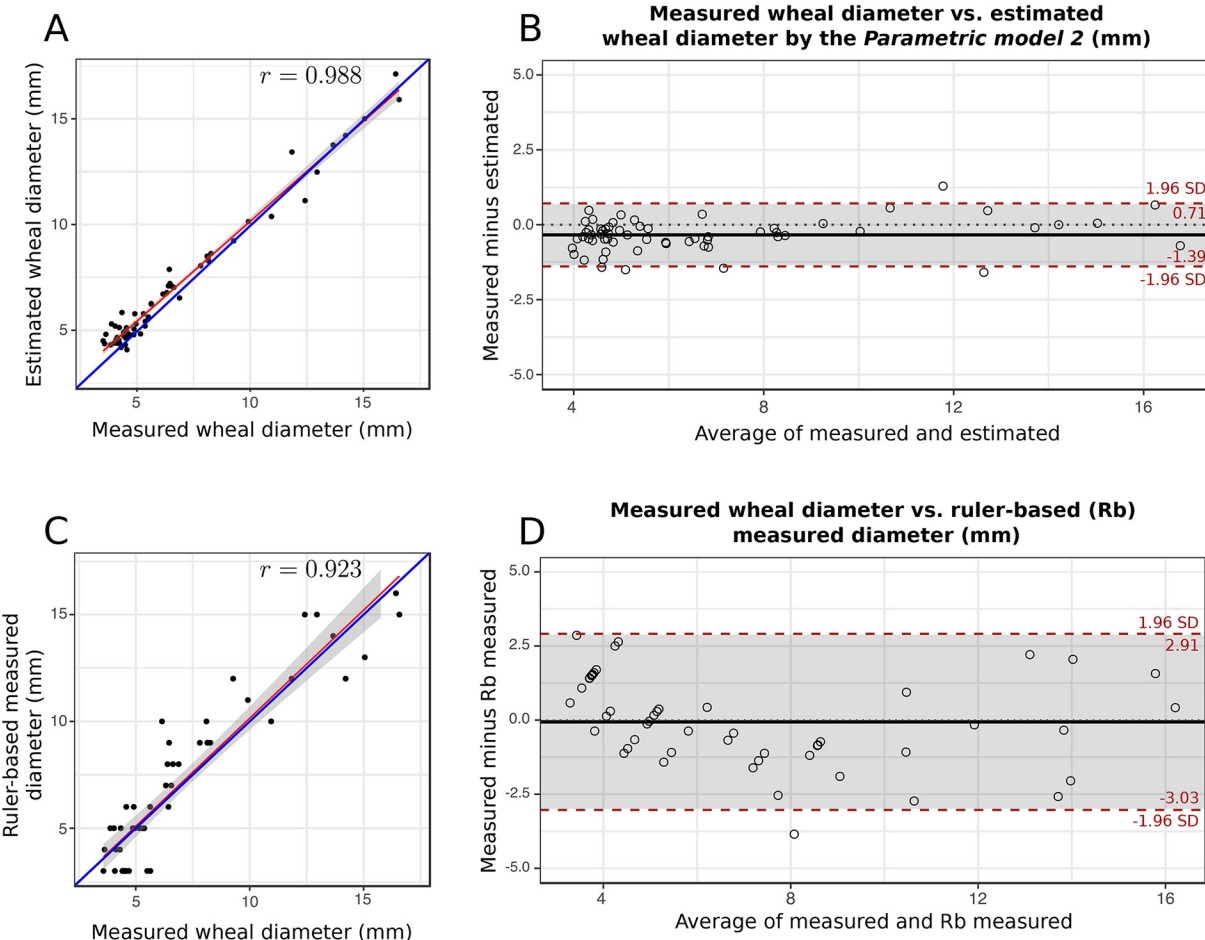

**Fig 9.** **(A)** Scatter plot, and **(B)** Bland-Altman plot of measured wheal diameter vs. estimated wheal diameter by the parametric model 2. The 95% agreement limits are [−1.58, 0.78] mm, which are close to the desired ±1 mm. **(C)** Scatter plot, and **(D)** Bland-Altman plot of measured wheal diameter vs. ruler-based measured diameter. Although, the mean difference is close to zero the 95% agreement limits are [−3.03, 2.91] mm, which are three times larger than the desired ±1 mm.

measurements of small wheals between 4 and 5 mm, where it is more likely that the methods differ. Nevertheless, the mean difference is sufficiently small for the diagnostic purposes of the SPT and the 95% agreement limits [−1.39, 0.71] mm are the closest to the desired ±1 mm.

The results from the correlation and Bland-Altman analyses show that both parametric models produce statistically similar results. However, the parametric model 2 on average yields diameters closer to the reference measurements (model 1: $\bar{d} = -0.398$ mm vs. model 2: $\bar{d} = -0.339$ mm) with narrower 95% limits of agreement (model 1: [−1.58, 0.78] mm vs. model 2: [−1.39, 0.71] mm). For this reason, we consider the parametric model 2 as our candidate de facto model for performing the wheal diameter estimation in our system, and the one chosen for graphical representation. In Fig 9(A) and 9(B), we show the regression line and the Bland-Altman plot between the manual reference measurements and estimated diameters by the parametric model 2, respectively.

**Accuracy of the ruler-based method.** The ratio of the manual reference measurements and the ruler-based measurements was 1.09 ± 0.132 or (0.96-1.22) with 99% confidence. The regression line between the manual reference measurements and the ruler-based measurements by the physician is shown in Fig 9(C). The regression equation $y = 1.022$(95%

CI = 0.899-1.146)$x$ – 0.099(95% CI = -1.080-0.881) and the Pearson's correlation coefficient $r$ = 0.923 [95% confidence interval, CI = 0.868-0.956, P ≪ 0.001] indicate a lower correlation than the obtained with the *Parametric model 2*. Additionally, in the Bland-Altman plot in Fig 9(D), we notice a wider distribution of the measurement differences compared to the proposed method. The mean difference $\bar{d} = -0.0604$ mm and the SD = ± 1.516 mm. Although, the mean difference is close to zero the 95% agreement limits are [−3.03, 2.91] mm, which are three times larger than the desired ±1 mm.

### Experiment 2

In Table 3, we show the estimated wheal diameters from the parametric model 2 for each of the five forearm positions as detected from the two cameras of the 3D imaging system. The last three columns show the mean, the SD, and the CV for each estimated wheal diameter. The physician measurement of each wheal with the millimeter ruler is reported. SPT negative reactions are marked with "—". Missing values, such as those for unobserved wheals for certain positions, are represented by "*". Note that the proposed method achieves highly consistent measurements. This observation is in agreement with the reported CV for each wheal. The highest CV obtained was 5.24% (*D. pteronyssinus*), which accounts for an SD of 0.28 mm and a mean diameter of 5.37 mm. The remaining CVs from the experiment were all below 5%, thus showing the high reproducibility of the proposed method. These variations are undetectable to the human eye, further confirming that the proposed wheal measurement method works reliably.

Moreover, the estimated wheal diameters are in overall agreement with the measurements reported by the physician, considering that the resolution of the ruler is 1 mm. Note that for two wheals (*C. dander* and *N. control*) the physician assigned a negative reaction (no wheal) probably because the wheal was small. However, the 3D imaging system has a high depth-sensitivity, and the wheals are correctly detected. It is not uncommon for false-positive reactions to appear at the site of the negative control due to the trauma that the pricking device imparts on the skin [43]. Moreover, we are certain of the detected wheals due to the regularly distributed prick test sites provided by the multi-test device, as shown in Fig 1(A). The measurement discrepancies may be due to the ruler resolution and irregular wheal shapes that by visual inspection could lead to significant variability.

### Automated SPT assessment time

The total runtime of the automated reading of the SPT depends on the number of wheal reactions. However, to give the reader an idea of the time the implemented system takes to scan

**Table 3. Results of the skin prick test.** Measurements are reported in millimeters.

| Allergen | Physician Measurement | Camera 1 | | | | | Camera 2 | | | | | Mean | SD | CV[%] |
|---|---|---|---|---|---|---|---|---|---|---|---|---|---|---|
| | | (1) | (2) | (3) | (4) | (5) | (1) | (2) | (3) | (4) | (5) | | | |
| *P. Americana* | 4 | 4.54 | * | 4.89 | 4.69 | 4.81 | * | 4.59 | * | * | * | 4.70 | 0.1466 | 3.12 |
| *D. farinae* | 11 | 10.19 | * | 10.58 | 10.14 | 10.08 | 10.39 | 10.59 | * | 10.20 | 10.14 | 10.29 | 0.2040 | 1.98 |
| *D. pteronyssinus* | 5 | * | * | 5.44 | * | 5.86 | 5.11 | 5.06 | 5.47 | 5.16 | 5.48 | 5.37 | 0.2813 | 5.24 |
| *B. tropicalis* | 10 | * | * | 6.89 | * | * | 6.83 | 6.58 | 6.71 | 6.42 | 6.23 | 6.61 | 0.2523 | 3.82 |
| *C. dander* | — | 4.36 | * | 4.69 | 4.61 | 4.25 | 4.39 | * | * | 4.21 | 4.31 | 4.40 | 0.1810 | 4.11 |
| *D. dander* | — | — | — | — | — | — | — | — | — | — | — | — | — | — |
| *P. control* | 8 | * | * | * | * | * | 7.24 | 7.10 | 7.03 | 7.09 | 7.22 | 7.14 | 0.0902 | 1.26 |
| *N. control* | — | * | * | * | * | * | 4.51 | * | 4.50 | 4.53 | 4.67 | 4.55 | 0.0793 | 1.74 |

and process the 3D data, we report here the time it took for the example shown in Fig 4. In this case, the subject's arm had four wheals. Our system was controlled via a PC with Windows 7 (2.4 GHz i7 intel processor, 8 GB RAM) and MATLAB R2017. The overall processing includes three stages: acquisition, 3D reconstruction, and robust diameter estimation. The acquisition of the images takes around 500 ms. This time is the same for all acquisitions regardless of the number of wheals. This stage is the only one in which the subject has to interact with the device. The 3D reconstruction stage includes the fringe image processing via Fourier Transform Profilometry and the phase-to-metric coordinate mapping. This stage takes around 1.5 s. Finally, the robust estimation of the wheal diameters requires about 14 s. This stage includes the pyramidal decomposition of the 3D reconstruction ($\sim 0.061$ s), the wheal detection ($\sim 4.56$ s), and the parametric fitting ($\sim 9.48$ s). The total runtime was 16 s. While this time could be further improved with parallel computing, we believe it is a reasonable processing time compared to 1.5 minutes on average that the physician takes to assess the SPT manually. However, further research is needed in a clinical setting to assess this aspect correctly. Moreover, if the physician requires a permanent record of the test, a traced copy of the wheals takes several minutes to produce. Nevertheless, the primary purpose of our approach is to have a digital record of the test with accurate and reliable wheal measurements.

## Discussion

Characterizing a wheal from the SPT with a millimeter-graded ruler is error prone [8]. To overcome this limitation the "scanned area" method was proposed [7], but it is too cumbersome to be carried out in the clinical practice. The 3D reconstruction of the SPT enables an unprecedented digital record of the test. Although, more importantly, the proposed models deliver accurate and reproducible measurements, and agree sufficiently well with the manual reference measurements from the physician. The CV between estimated diameters was 6.40%, which exhibits high reproducibility between the estimations from both models. However, the parametric model 2 on average yields diameters closer to the reference measurements (model 1: $\bar{d} = -0.398$ mm vs. model 2: $\bar{d} = -0.339$ mm) with narrower 95% limits of agreement (model 1: $[-1.58, 0.78]$ mm vs. model 2: $[-1.39, 0.71]$ mm), which are close to the desired ±1 mm. The average difference of -0.330 mm is not altogether a problem since it could be reduced through calibration to ensure an average difference closer to zero with training data. This flexibility and degree of accuracy cannot be achieved with the conventional methods, or even with the automated scanned area methods. Nevertheless, this recalibration procedure merits further investigation. The agreement is with respect to manual reference measurements from a physician, but as we showed at the beginning of this paper, our system has sub-millimeter resolution and accuracy, which make the automated reading of the SPT much more reliable. Moreover, our proposed method for automatically and robustly estimating the wheal diameter produces accurate and reproducible results with CVs of under 5%.

## Conclusion

In this work, we showed that the wheals from the skin prick test could be digitized in 3D and further processed to obtain accurate and reliable measurements. Our proposed method takes the 3D surface of the forearm as input, removes the global surface to isolate the wheals, automatically detects them, and performs a robust parametric fitting to produce an estimation of the wheal longest diameter. We proposed two parametric models inspired by the two conventional approaches for the diameter estimation in the assessment of the SPT. Through experimental results, we showed that the proposed models provide accurate and reliable measurements, and agree sufficiently well with manual measurements from the physician.

Moreover, the proposed method delivers reproducible results even by moving the arm of a subject through different positions.

## Acknowledgments

The authors would like to thank the volunteering subjects (primarily from the Universidad Tecnológica de Bolívar and Universidad de Cartagena), as well as Hernando Altamar and Andrés Merlano for their help in experimental tests. J. Pineda and R. Vargas thank Universidad Tecnológica de Bolívar for a post-graduate scholarship.

## Author Contributions

**Conceptualization:** Lenny A. Romero, Javier Marrugo, Jaime Meneses, Andres G. Marrugo.

**Funding acquisition:** Lenny A. Romero, Andres G. Marrugo.

**Investigation:** Jesus Pineda, Raul Vargas, Javier Marrugo.

**Methodology:** Lenny A. Romero, Javier Marrugo, Jaime Meneses, Andres G. Marrugo.

**Project administration:** Lenny A. Romero, Andres G. Marrugo.

**Resources:** Andres G. Marrugo.

**Software:** Jesus Pineda, Raul Vargas.

**Supervision:** Lenny A. Romero, Andres G. Marrugo.

**Visualization:** Jesus Pineda, Raul Vargas.

**Writing – original draft:** Jesus Pineda, Andres G. Marrugo.

**Writing – review & editing:** Lenny A. Romero, Javier Marrugo, Andres G. Marrugo.

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
