## [Decision Letter · Decision Letter 0]

3 Jul 2019

PONE-D-19-17003

Robust automated reading of the skin prick test via 3D imaging and parametric surface fitting

PLOS ONE

Dear Dr. Marrugo,

Thank you for submitting your manuscript to PLOS ONE. After careful consideration, we feel that it has merit but does not fully meet PLOS ONE’s publication criteria as it currently stands. Therefore, we invite you to submit a revised version of the manuscript that addresses the points raised during the review process.

We would appreciate receiving your revised manuscript by Aug 17 2019 11:59PM. To enhance the reproducibility of your results, we recommend that if applicable you deposit your laboratory protocols in protocols.io, where a protocol can be assigned its own identifier (DOI) such that it can be cited independently in the future. For instructions see: http://journals.plos.org/plosone/s/submission-guidelines#loc-laboratory-protocols

We look forward to receiving your revised manuscript.

Kind regards,

Enrico Grisan

Academic Editor

PLOS ONE

Journal Requirements:

2.

We note that you have indicated that data from this study are available upon request. PLOS only allows data to be available upon request if there are legal or ethical restrictions on sharing data publicly. For more information on unacceptable data access restrictions, please see http://journals.plos.org/plosone/s/data-availability#loc-unacceptable-data-access-restrictions.

Additional Editor Comments (if provided):

Dear Dr. Marrugo and dear co-authors,

your manuscript has been reviewed by referees and by myself, and the comments agree that you manuscript and your proposed methods is interesting and has merits.

However, there are some aspects of the work and of the results that need to be improved and clarified before the manuscript can be accepted.

I hope that the comments and your revision will make a stronger case for your work and widen the interest when published.

With best wishes

Enrico Grisan

Reviewers' comments:

Reviewer's Responses to Questions

**Comments to the Author**

1. Is the manuscript technically sound, and do the data support the conclusions?

Reviewer #1: Partly

Reviewer #2: Yes

2. Has the statistical analysis been performed appropriately and rigorously? 

Reviewer #1: Yes

Reviewer #2: Yes

3. Have the authors made all data underlying the findings in their manuscript fully available?

Reviewer #1: No

Reviewer #2: Yes

4. Is the manuscript presented in an intelligible fashion and written in standard English?

Reviewer #1: Yes

Reviewer #2: Yes

5. Review Comments to the Author

Reviewer #1: In my opinion, there are two serious objections about the technical quality of this work.

1) The authors claim that, to the best of their knowledge, this is the first time such a model (a parametric model to characterizing the 3D data of the wheals) is proposed (contribution in page 3). This model is shown in equation (1), that use Gaussian functions with a flat-top. In principle, the model sounds well, but if it the first time that a model is proposed, the authors should justify better the selection of the model. For example, take three or four possible models and test which one achieves better fit.

Related to this issue, I would like to know if Fig. 3 shows real results of the proposed method. I mean, if you compare the images of S’_w1, S’_w2, etc. with f_1, f_2, etc., the obtained model does not represent well the actual wheal. The shape of the “base” of the resulting model seems to be rectangular and I would like to use a model that achieves a kind of elliptical base. The use of Gaussian functions could achieve this, but I think that the use of super-Gaussian functions or Gaussian functions with a flat-top provokes that the base of the modeled wheal is nearly rectangular. In my opinion, it has no sense to model very precisely the high of the wheal but the base of the wheal (moreover if you want to obtain the maximum diameter of this base). Maybe a kind of cylindrical function, using an ellipse for the base and a constant high is a better model for this purpose.

Thus, if proposing a model is one of the main contributions of the paper, I would like to discuss better the goodness of the model.

2) The authors admit in page 5 that S’ includes bumps due to wheals but also pores, follicles, wrinkles and hair (that are not wheals but the algorithm could see as wheals). However, in page 6 section wheal detection, they assume that applying a blob detector the wheals are automatically detected, and cite reference [27]. Are you completely sure that you are not obtained false wheals? The authors mention in the introduction, page 2, that a previous work has coped with false positives [13]. This issue should be better discussed. In fact, I’m not very convinced that the 11 wheals not detected by the physician through visual inspection are real wheals (page 9). Moreover, Table 3 reports real wheal for both the positive control (histamine) and the negative control (diluent !). For the positive control, a wheal is expected… but, for the negative control no wheal should arise!

Read please again the sentence “Note that for two wheals (cat dander and negative control) the physician assigned a negative reaction (no wheal) probably because the wheal was very small” (page 11). It is very strange. Probably the physician is right, and at least with the negative control (diluent) there was no wheal (and the pretended wheal reported in Table 3 is a false positive). The authors must clarify this point.

The work is promising, but I encourage the authors to work in these two points before publishing it.

Reviewer #2: In the manuscript, the authors present an excellent report on the performance of a new 3D imaging system they have developed for automated parameterizing of wheals caused by allergic reactions during the skin prick test commonly used in medicine. They go in considerable detail to explain nearly every aspect of its construction and operation, including the underlying mathematical operations that automatically transform a clinical image of a patients skin into wheal diameter information. They include a clinical study in which they compare the performance of their system to standard methods of assessing allergic reactions as used by medical specialists in diagnoses and convincingly show a very promising potential of the new method.

There are, in my opinion, a few minor details that, when resolved, would improve the presentational value and consistency of the manuscript. They are (in the order as they appear in the manuscript):

1. In the Introduction section, the authors might be interested in the paper (Laloš, Mrak, Pavlovčič, Jezeršek, Handheld optical system for skin topography measurement using Fourier transform profilometry, Journal of Mechanical Engineering 61(5), pp. 285-291 (2015), doi: 10.5545/sv-jme.2015.2424) in which similar pattern projection and triangulation methods have been used to determine the skin surface topography.

2. In the Contribution section, lines 79 and 80, the authors successively reference Fig. 1 and Fig. 3. It is unusual to skip a number when first referencing figures. They might consider referencing Fig. 2 in between the two to maintain the proper figure numbering.

3. In Tables 1 and 2, the authors use Greek mu and sigma signs to denote average and standard deviation, respectively. While this is a common practice, explicitly stating what these symbols represent, by adding them into text in lines 130 and 131, for example, would preserve the consistency of the manuscript.

4. It would be most illuminating if the authors could add an unaltered image of the patients forearm, such as what a physician would normally see during his evaluation, to Fig. 4 so that it could be compared to their images denoted S and S'. Provided such an image exists at all.

5. In Eq. (1), the left curly bracket is used very unusually. It might be more elegant to separate the equations and give each of them their own number.

6. It is not apparent how the rotational angle theta in the second part of Eq. (1) and in the line 190 is obtained. It would be good to explain this.

7. It might be an informative duscussion to briefly compare the times needed to arrive to the wheal diameters by the new metod with those needed to obtain such diameters by standard manual methods.

In view of this, I do recommend the manuscript entitled 'Robust automated reading of the skin prick test via 3D imaging and parametric surface fitting' to be published in the PLOS ONE journal with minor revisions.

6. PLOS authors have the option to publish the peer review history of their article (what does this mean?). If published, this will include your full peer review and any attached files.

Reviewer #1: No

Reviewer #2: No

---

## [Author Response · Author response to Decision Letter 0]

20 Aug 2019

The response to reviewers is included as an attachment.

---

## [Decision Letter · Decision Letter 1]

25 Sep 2019

Robust automated reading of the skin prick test via 3D imaging and parametric surface fitting

PONE-D-19-17003R1

Dear Dr. Marrugo,

We are pleased to inform you that your manuscript has been judged scientifically suitable for publication and will be formally accepted for publication once it complies with all outstanding technical requirements.

With kind regards,

Enrico Grisan

Academic Editor

PLOS ONE

Additional Editor Comments (optional):

Reviewers' comments:

Reviewer's Responses to Questions

**Comments to the Author**

1. If the authors have adequately addressed your comments raised in a previous round of review and you feel that this manuscript is now acceptable for publication, you may indicate that here to bypass the “Comments to the Author” section, enter your conflict of interest statement in the “Confidential to Editor” section, and submit your "Accept" recommendation.

Reviewer #1: All comments have been addressed

Reviewer #2: All comments have been addressed

2. Is the manuscript technically sound, and do the data support the conclusions?

Reviewer #1: Yes

Reviewer #2: Yes

3. Has the statistical analysis been performed appropriately and rigorously? 

Reviewer #1: Yes

Reviewer #2: Yes

4. Have the authors made all data underlying the findings in their manuscript fully available?

Reviewer #1: No

Reviewer #2: Yes

5. Is the manuscript presented in an intelligible fashion and written in standard English?

Reviewer #1: Yes

Reviewer #2: Yes

6. Review Comments to the Author

Reviewer #1: Thank you for your responses and the revised manuscript. I think that the sections “discussion” and “conclusion” could be merged and better written, but the work is acceptable for publication.

Some minor details:

1) In the abstract there is an errata: “and we propose a we propose a general…”

2) Use “skin prick test” (abstract) or “Skin prick test” (introduction) always in the same form, without or with capitals, but do not use the two forms.

3) Fig. 4 and Fig. 5 are the same in the manuscript, but it seems that the correct Fig. 5 is included in page 30 of the PDF of the submission. Use the correct Fig. 5 in the final manuscript.

4) Page 9 equation (1), I would have written cos theta and sin theta, that is, without parenthesis cos(theta) and sin(theta).

5) Page 10 Subjects and test, for the age I read “18-60 years” in the first line and “18-48 years” in the third line. I wonder if it is an errata.

6) Page 13 and page 14. Maybe instead of simply “Case 1” and “Case 2” I would have said “case 1 of Fig. 8” or “case 2 in Fig. 8”.

7) Page 13 and page 14, for “parametric model 1” and “parametric model 2” use italic or regular, but not both forms.

Reviewer #2: I feel that my comments have been successfully addressed and the overal quality of the manuscript has been considerably improved. Therefore, I do recommend the revised manuscript for publication.

7. PLOS authors have the option to publish the peer review history of their article (what does this mean?). If published, this will include your full peer review and any attached files.

Reviewer #1: No

Reviewer #2: No

---

## [Editor Report · Acceptance letter]

8 Oct 2019

PONE-D-19-17003R1 

Robust automated reading of the skin prick test via 3D imaging and parametric surface fitting 

Dear Dr. Marrugo:

I am pleased to inform you that your manuscript has been deemed suitable for publication in PLOS ONE. Congratulations! Your manuscript is now with our production department. 

With kind regards,

on behalf of

Dr. Enrico Grisan 

Academic Editor

PLOS ONE